# The Sleep Impact on Activity Diary (SIAD): A Novel Assessment of Daytime Functioning in Insomnia

**DOI:** 10.3390/brainsci11020219

**Published:** 2021-02-11

**Authors:** Kelsey Bickley, Nicole Lovato, Leon Lack

**Affiliations:** 1CRC for Alertness, Safety and Productivity, Notting Hill, VIC 3168, Australia; nicole.lovato@flinders.edu.au (N.L.); leon.lack@flinders.edu.au (L.L.); 2The Adelaide Institute for Sleep Health (AISH), a Flinders University Research Institute, Bedford Park, SA 5042, Australia; 3College of Education, Psychology, and Social Work, Flinders University, Bedford Park, SA 5042, Australia

**Keywords:** insomnia, daytime functioning, impairments, assessment, task participation, effort

## Abstract

Daytime impairments feature in the diagnostic criteria for insomnia disorder yet are rarely assessed comprehensively in clinical practice and tend to focus on mood and subjective assessment of cognitive competence. In order to gain more information about the engagement in daily activities we developed the Sleep Impact on Activity Diary (SIAD). This initial investigation included 22 insomnia patients (15 females, aged 49.9 years, SD = 17.6) and 19 normal sleeper controls (13 females, aged 30.9 years, SD = 8.9). For 14 consecutive evenings, participants rated how their prior night-time sleep impacted their participation in 12 common daytime activities (e.g., work, self-care, leisure). They also rated how much effort each activity required (Range: 0–4). Overall, insomnia patients participated in only one fewer activity type per day (*M* = 7.48, *SD* = 1.34) than controls (*M* = 8.39, *SD* = 1.43) (*p* = 0.041, *d* = 0.66). More noteworthy, they reported that sleep negatively affected their participation more than controls (*M* = 1.56, *SD* = 0.92 versus *M* = 0.23, *SD* = 0.35; *p* = < 0.001, *d* = 1.90), and that activities required more effort (M = 1.58, SD = 0.64 versus M = 0.81, SD = 0.76; *p* = 0.001, *d* = 1.10). This pilot study with the SIAD suggests that, compared to good sleepers, insomnia patients participate in somewhat fewer activities but that their activities require considerably more effort and are adversely affected by their sleep. The SIAD tool promises to provide a more comprehensive picture of the everyday impact of insomnia. It remains to be validated on a much larger sample in a clinical treatment study.

## 1. Introduction

Symptoms of poor daytime functioning are a core component of chronic insomnia diagnoses [1,2]. These criteria are exceptionally broad, including feelings of fatigue or low energy, daytime sleepiness, impaired attention, concentration or memory, mood disturbances, behavioral difficulties, and impaired occupational, academic, interpersonal, or social functioning [1,2]. Compared to normal sleepers, insomnia patients frequently complain of low energy, poor mood, and difficulties with concentration [3]. These subjective impairments, especially low energy/fatigue, are so severe that they are often the prompt for individuals to seek treatment [4,5]. Insomnia patients also perform worse than normal sleepers on objective memory, perception, and problem-solving tasks [6,7]. However, not all studies have found lower objective performance in cognitive tasks (e.g., divided and sustained attention, working memory) [6,7]. This suggests a disparity between objective measures of cognitive performance, usually from experimental studies, and the subjective reports of negatively impaired daytime functioning. It may be the case that the relatively brief and challenging cognitive tasks in the laboratory environment may not be sensitive enough to pick up subtle impairments. Alternatively, insomnia patients may be able to rally the effort to perform normally in these conditions but not be able to sustain this effort throughout more extended normal daily conditions. In the attempt to develop greater ecological validity in measures of the reported impairments associated with insomnia, we developed the Sleep Impact on Activity Diary (SIAD). The SIAD aims to measure the pragmatic daytime functioning of individuals with respect to the types of activities engaged, whether the activity was negatively affected by their sleep and how much effort they felt was required in carrying out the activity. This may provide additional, useful information to clinically assess the consequences of poor sleep in those with insomnia and to monitor the daytime functioning benefits of insomnia treatment.

The SIAD requires individuals to report on twelve common daily activities including self-care, leisure, work, and social engagement. Existing questionnaires like those that measure Health Related Quality of Life (HRQoL) also tap into activity participation and describe functioning deficits in the insomnia population [8,9,10]. However, most of these questionnaires require judgements for a retrospective period of time and are susceptible to poor or biased memory. As sleep diaries are considered more reliable measures of sleep than retrospective question of typical or average sleep, the SIAD is administered daily and presumably less impacted by memory-related biases that would be problematic in the arguably memory-impaired insomnia population. One exception to retrospective questionnaires is The Daytime Insomnia Symptom Scale (DISS) that collects daily assessments from participants mainly relating to mood, alertness, and fatigue [11]. Daily assessments of mood have demonstrated sensitivity to changes in sleep [12] and are important aspects of the negative consequences of insomnia. However, another important aspect of daytime functioning that has hitherto been neglected in diary measures is that of assessing the engagement in normal daytime activities. The SIAD aims to assess this engagement in addition to judgements of the extent to which the engagement in those activities was negatively impacted by prior poor sleep. Regarding the inclusion of effort assessments, several authors have proposed that insomnia patients may perform at normal levels in short-duration cognitive tasks by putting in more effort than controls [13,14,15]. We propose that they would attempt to do the same in daytime activities. However, this additional effort will not always be summoned and thus result in impaired performance. Ultimately, both the negative impact of sleep and effort to participate could be important indicators of daytime functioning deficits in the insomnia population.

Overall, we expect group differences in all three measures of the SIAD. Aligning with the HRQoL deficit reported in the insomnia population [8,9,10] we expect that the insomnia group will report engagement in fewer activities than good sleepers and for the activities in which they do engage they will report more negative impact from their prior sleep and greater perceived effort associated with engagement.

## 2. Materials and Methods

### 2.1. Participants

This investigation included 41 individuals, consisting of 22 chronic insomnia patients (15 = female, 7 = male) with a mean age of 49.91 years (SD = 17.59) and 19 normal sleepers (13 = female, 6 = male) with a mean age of 31 years (SD = 8.90). Insomnia participants were incoming patients at the Adelaide Institute for Sleep Health (AISH) insomnia treatment clinic prior to any treatment. Normal sleeper participants were recruited from paper advertisements at the Flinders University campus in Bedford Park, Adelaide, and via word of mouth. All recruitment occurred between August 2015 and May 2017. The Ethics Review Committee (RPAH Zone) of the Sydney Local Health District approved this project (project number: X14-0389 & HREC/14/RPAH/517).

The eligibility criteria for the insomnia group included a diagnosis of Insomnia Disorder following the ICSD-III and DSM-V criteria for chronic insomnia [1,2] as confirmed by a sleep physician and an Insomnia Severity Index (ISI) score ≥ 10 [16]. The control group were required to have stable, healthy sleep, i.e., no difficulties initiating and/or maintaining sleep, no daytime impairments, and an ISI score of <10.

Exclusion criteria for all participants included: Under 18 years of age, not fluent in reading/writing English, cognitive impairments preventing individuals from understanding study directions, pregnancy or current lactation, active use of illicit substances, significant caffeine or alcohol (>4 standard drinks daily) dependence, worked nightshift between 12 am and 3 am in the last 6-months, or travelled across more than 3 time zones in the last 2-months.

### 2.2. Measures

#### 2.2.1. Insomnia Severity Index (ISI)

The ISI is a 7-item questionnaire that assesses sleep quality and daytime functioning. Responses are collected on a 5-point Likert scale. Higher numbers indicate increased severity, and a global score comprising all of the items indicates overall severity of insomnia symptoms. Evaluations of the ISI demonstrate that is it an effective screening tool for patients with insomnia complaints with strong internal reliability (Cronbach *a* = 0.90–0.91) [16,17]. In our study we used a cut-off score of 10 as previous work had determined this to be optimal (sensitivity = 86.1%, specificity = 87.7) for the detection of insomnia in community samples [16].

#### 2.2.2. Sleep Impact on Activity Diary (SIAD)

The SIAD contains twelve common daily activities, see Figure 1. These included basic activities necessary for independent living (e.g., grooming, eating meals, etc.) used for assessing activities of daily living in the elderly [18]. These basic activities were expanded to include common activities of adults across the whole age range (e.g., work/volunteering/education, care for children, exercise, leisure activities). Each activity engaged in scored 1, with a maximum sum of 12. Additionally, for each of the 12 activities in which they participated they reported: (1) How their sleep on the previous night negatively affected their ability to participate (responses ranging from: 0 = ‘not at all’ to 4 = ‘a great deal’), and (2) how much effort it took to participate in the activity (responses ranging from: 0 = ‘no effort’, up to 4 = ‘extreme effort’). In cases where participants had completed the same activity multiple times throughout the day, they indicated the average amount of impact of their sleep and their effort for that activity. This resulted in a single sleep impact and effort value for each activity on each day. Activities that participants did not participate in each day were indicated by reporting ‘N/A’ and not scored.

#### 2.2.3. Morning Consensus Sleep Diary (CSD-M)

Participants were required to complete a paper copy of the morning version of the Consensus Sleep Diary (CSD-M) [19]. Every morning, they reported their nocturnal sleep-related data including ‘go to bed’ time, attempt to sleep time, estimated sleep onset latency (SOL), estimated wake after sleep onset (WASO), final wakeup time, and ‘out of bed’ time. From these, total sleep time (TST), time in bed (TIB), and sleep efficiency (SE) were calculated.

### 2.3. Procedure

Preliminary eligibility to participate was determined during a 15-min telephone-screening questionnaire following which all potential participants filled out an online version of the ISI. Potential insomnia participants then attended a clinical interview with a sleep physician at the AISH sleep clinic who ensured that they met the diagnostic criteria for Insomnia Disorder. Normal sleeper participants went straight into the study on the basis of the telephone questionnaire.

Eligible participants completed a paper copy of the SIAD every evening, reflecting back on the activities they had completed that day. Participants were encouraged to do this just before their bedtime so they could include all activities that they participated in that day. In the morning, within thirty minutes of awaking, participants completed their sleep diary regarding their previous night’s sleep. Participants completed these assessments in their home environment for two weeks.

### 2.4. Statistical Analysis

#### 2.4.1. Missing Data

Regarding the CSD-M, in the insomnia group 16 individuals provided all 14-days of data, 2 individuals each provided 13, 11, and 10-days of data (5.2% missing data). In the control group 15 individuals provided 14-days of data, 1 individual provided 13 and 7- days of data, and 2 individuals provided 11-days of data (5.2% missing data). Regarding the SIAD, in the insomnia group 15 individuals provided all 14-days of data, 3 individuals provided 13-days of data, 3 individuals provided 11-days of data, and 1 individual only provided 7-days of data (6.2% missing data). In the control group 14 individuals provided 14-days of data, 2 individuals provided 13-days of data, 1 individual provided 12, 10, and 7-days of data (5.6% missing data). It was felt that the low percentage of missing data comparable between the two groups did not justify statistical compensation for missing data.

#### 2.4.2. Analysis Procedure

We conducted group comparisons with SPSS Statistics Version 22.0 (IBM Corp. Released 2013. Armonk, NY, USA). All group comparisons were made with Chi-square and Independent Samples T-Tests. Group demographic and sleep characteristics means were compared first (Table 1). To determine if patients with insomnia performed a similar number of activities compared to controls, we calculated the first of three global SIAD scores. That is, the average daily count of the 12 activities that were participated in per individual. To assess if the insomnia group reported a greater negative impact of sleep on the activities they participated in and required more effort to do so than controls, the remaining two global scores were calculated. These were represented by means that included one average daily negative sleep impact and one effort score per individual for all participated activities. That is, each individual contributed one sleep impact and one effort score per day for approximately 14-days with the daily means for each group reported below in Table 2 and Table 3.

## 3. Results

### 3.1. Participant Demographics and Sleep Characteristics

Patient demographic (age and ISI) and sleep characteristics (CSD-M) were calculated and presented as per convention [16,17,19] (Table 1). There was no difference in gender ratios between groups *X*^2^ (1, *n* = 41) = 0.000, *p* = 0.987. The insomnia group was older than the control group and reported greater ISI scores than controls. Table 1 also contains the sleep-related outcomes from the sleep diary during the two-weeks of data collection. Consistent with higher ISI scores the insomnia group reported greater SOL and WASO durations than controls. The insomnia group also reported shorter TST and lower SE than controls, despite no significant differences in attempting to sleep and getting out of bed times, and subsequently no difference of TIB (from “Bedtime” to “OutBed”).

### 3.2. Global SIAD Scores

Insomnia patients reported participating in about one fewer activity per day (*M* = 7.48, *SD* = 1.34) than controls (*M* = 8.39, *SD* = 1.43) (*M Difference =* −0.91) *t* = (39) −2.111, *p* = 0.041, *d* = 0.66 (Figure 2).

Individuals in the insomnia group reported their participation as much more negatively affected by their sleep (*M* = 1.56, *SD* = 0.92) than did the control group (*M* = 0.23, *SD* = 0.35), *t* = (27.65) 6.241, *p* < 0.001, *d* = 1.90. The insomnia group also reported that it took more effort to participate in activities *M* = 1.58 [*SD* = 0.64] compared to the control group *M* = 0.81 [*SD* = 0.76] (*t* = (39) 3.518, *p* = 0.001, *d* = 1.10) (Figure 3).

### 3.3. SIAD Scores for Sleep Impact on Participation in Specific Activities

Regarding individual activities a greater negative impact of sleep was reported by the insomnia group for almost all activities across 14-days with moderate to large effect sizes. For the remaining item #6 (attended school/studied) the marginal significance was due to greater variability in the insomnia group and lower degrees of freedom.

### 3.4. SIAD Scores for Effort to Participate in Activities

Regarding the mean effort scores for individual activities, only activity #3 (health tasks) and #4 (eating meals) did not show more effort reported by the insomnia group (Table 3). These two activities reported the least amount of effort and perhaps the truncated range of effort left less room for difference between groups.

## 4. Discussion

The aim of this pilot study was to investigate whether the Sleep Impact on Activity Diary (SIAD) showed differences between insomnia patients and normal sleepers in daytime activities in terms of number of activities participated, negative impact of sleep on the activities, and perceived effort of participation. In this investigation, the insomnia group performed about one fewer activity on an average day than the control group. This significant (*p* = 0.041) but medium effect size difference could have arisen mainly from item #6 (School/studied). The recruitment process resulted in a younger control group most of whom (68%) happened to be university students while only 18% of the insomnia group were participating in education. Of the other eleven possible activities, participation rate was essentially equal between the insomnia group (86%) and control group (88%). Since participation in educational activities is a long term commitment, it is less of a voluntary choice from day to day and less likely to be a function of how motivated the individual may feel on the day. Future research comparing insomnia patients and good sleepers from the same demographic is necessary to confirm if there is a difference in the number of activities engaged.

The most striking difference between groups was that the insomnia patients reported virtually all the activities as more negatively affected by their sleep than the controls. At present the SIAD does not have questions to elicit information about the ways in which participants feel their activities have been adversely affected. Our presumption has been that a “negative impact” implies that the activity was not performed as well as it should have been in terms of quantity and/or quality. Future iterations of the SIAD could include further questions to make their perceived negative impacts more explicit. However, this would make the questionnaire more time consuming and possibly result in lower response compliance. In any case, in this pilot study the insomnia patients felt their sleep had considerably more negative impact on engagement in virtually all of the activities.

The second most striking difference between the two groups was the perceived amount of effort involved in participating in these activities. Elevated fatigue, not sleepiness, is probably the most commonly reported daytime complaint expressed by insomnia patients and the impairment that most often motivates them to seek treatment. Fatigue tends to inhibit behavior. However, this inhibition can be overcome with greater effort. Perhaps this finding of greater effort to engage in activities reflects the greater amount of experienced fatigue throughout the day. What the findings of this study seem to be suggesting is that the insomnia patients engage in basically the same number of activities, perhaps as a result of social, familial, or work obligations. However, it is more difficult for them to do so requiring more effort on most of the activities. Despite this attempt to meet these obligations through more expended effort, their evaluation is that they still fall short of performing these obligations at an acceptable level.

Whether these subjective evaluations of their impaired performance on these everyday activities is objectively correct or more a reflection of their perceived effort is yet to be determined. It would seem plausible that in the context of feelings of fatigue and the need to extend more effort to perform activities (physical, mental, emotional) that the required effort to perform at desired levels of competence will often not be mustered, resulting in impaired performance. On the other hand, the insomnia patients report greater effort to participate in these daily activities and this greater effort may be compensating for feelings of fatigue resulting in normal competence in these activities. Methods of objectively and unobtrusively measuring competency in the performance of everyday activities are needed to confirm whether insomnia patients are objectively impaired and/or are underestimating their competence perhaps in response to feelings of fatigue and effort required in these activities. For example, Fabbri and colleagues used actigraphy to test prospective memory between insomnia patients and controls but, interestingly, found the insomnia patients to perform equally to the controls [20]. This is an area ripe for investigation. The purpose of developing the SIAD was to approach this aim by gathering more information about day-to-day activities and the impact of sleep on these activities. The results of this initial study suggest that insomnia patients engage in virtually the same number of different activities but feel more effort is required in their engagement and, despite this greater perceived effort, their activity is still impaired by their sleep.

### 4.1. Strengths and Limitations

Based on this initial evaluation of the SIAD, we propose that daytime activity and perception assessments are a viable and comprehensive tool for measuring daytime functioning in the insomnia population. A strength of this daily evaluation is that it is prospective rather than retrospective and thus less vulnerable to both memory loss and potential memory bias to the more negative experiences from the recent past. The SIAD shares this momentary assessment methodology with the Daytime Insomnia Symptom Scale [11] but extends the information beyond mood and alertness feelings to a full range of common daily activities. Basic activities of daily living which are often covered by existing HRQoL questionnaires, describe basic tasks including activities like eating, bathing, and leaving the house. These are often disturbed in physically impaired populations, e.g., individuals who are unable to feed themselves due to mobility paralysis [21]. The SIAD includes these basic activities in addition to more complex daily activities such as remote and face-to-face social contact and physical exercise. This allows us to measure the more subtle effects of insomnia disorder on the most commonly engaged activities across the total age span. Although SIAD assessments are daily, they only require a very small amount of time to complete at the end of the day when some reflective time is usually available. The SIAD is a simple tool that can easily be used alongside the daily sleep diaries that research and clinical insomnia patients already complete.

A major limitation of this pilot study was the demographic differences between the groups with the insomnia patients being older and less likely to be engaged in educational activities. It could be argued that perceived effort to engage in daily activities increases with age and thus confounded the effort measure. In addition there is the possibility that age increases the perceived negative impact of poor sleep on these activities. Although the control group was a sample “of convenience”, future comparisons between good sleepers and insomnia patients would benefit from a more careful matching of demographics between groups.

Another potential limitation of the present SIAD tool is that it does not go far enough to detail the number of times energy was expended, or the amount of time spent in each activity. An activity type engaged in once for a short time (e.g., single, brief walking exercise) counts the same as an activity engaged in several times for much longer (e.g., a few bouts of heavy exercise over a longer period of total time). However, the latter amount of detail would have required considerably more time to document in a questionnaire and most probably would have reduced compliance and increased the percentage of missing data. We tried to strike a reasonable compromise between capturing additional information of daily activities without requiring so much information to seriously degrade compliance with increasing missing data. Although there was some missing data in the present study, its small percentage (~5%) suggested to us that the SIAD struck the right balance between maximizing information while minimizing onerousness and missing data.

Although the global scores of number of activities, average impact, and average effort for all undertaken activities by an individual are of importance, they have the potential limitation of masking differences between types of activities undertaken. However, it is possible to analyze the activities separately for both impact and effort (such as in Table 3) thus expanding the detailed effects of differences between individuals or effects of treatment and the wealth of information for clinical researchers.

### 4.2. Future Directions

The SIAD is the first foray into documenting daily activities, the effort in this engagement and the perceived impact of poor sleep on this engagement. Future studies should: 1. Compare insomnia patients with matched controls across the total adult age range, and 2. Validate the SIAD against other daily measures such as the DISS or general levels of activity measured objectively such as with actigraphy. Although it was beyond the scope of this study with only 14 days of data, it would be important to investigate intra-individual day-to-day variation in sleep measures (both from the social variation between weekday and weekend and in spontaneous variation) with variation in number of activities, negative impact on activities, and required effort in activities. In the future, we believe that the SIAD could be used to assess fluctuations in daytime impairments, characteristic of insomnia disorder, both as an ongoing monitoring tool of impairment severity and to assess changes following treatment. In other words, is the SIAD responsive to treatment (e.g., less negative impact and less effort) as are other retrospective assessments of daytime functioning and mood?

## 5. Conclusions

This investigation demonstrated the potential of using the SIAD to measure impairments in daytime functioning in the insomnia population that current questionnaires do not comprehensively assess. The findings from this study may also explain some of the discrepancy between the subjective and objective assessment of daytime impairments for insomnia patients, suggesting that the perception of the negative impact of sleep and the effort required for each activity may impact perceived and actual performance. The SIAD may benefit the clinical treatment and management of insomnia by comprehensively assessing the severity of impairment and response to treatment.

## Figures and Tables

**Figure 1 brainsci-11-00219-f001:**
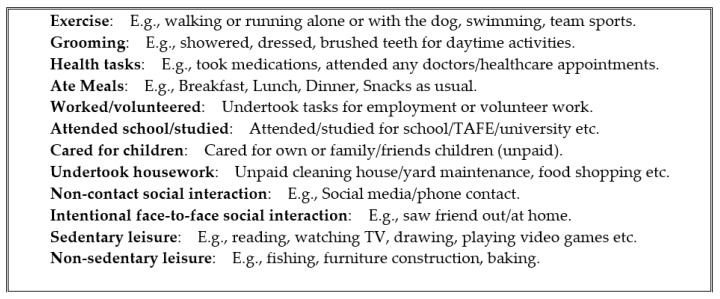
Daytime activity options and examples from the Sleep Impact on Activity Diary (SIAD).

**Figure 2 brainsci-11-00219-f002:**
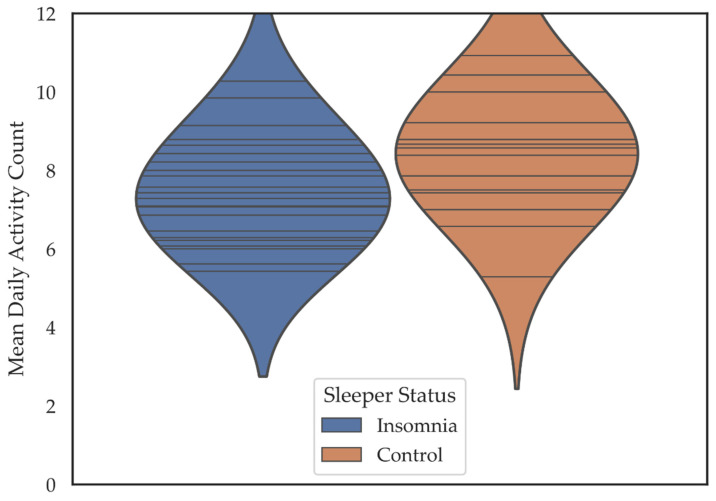
Violin plots depicting the mean daily activity participation count for individuals in the insomnia and control groups. Horizontal lines are individual data points. Colored regions are the Gaussian kernel density estimates (KDE) with bandwidth equal to one standard deviation of the data.

**Figure 3 brainsci-11-00219-f003:**
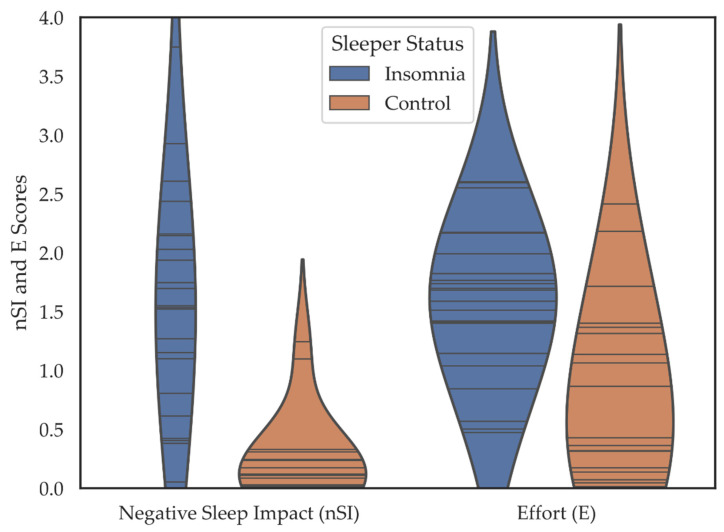
Violin plots depicting the negative sleep impact (nSI) and effort (E) scores including all 12 SIAD items over 14- days for individuals in the insomnia and control groups. Horizontal lines are individual data points. Colored regions are the Gaussian kernel density estimates (KDE) with bandwidth equal to one standard deviation of the data.

**Table 1 brainsci-11-00219-t001:** Participant demographics and sleep characteristics between the insomnia (*n* = 22) and control (*n* = 19) groups.

Variable	Units	Insomnia Mean (SD)	Control Mean (SD)	*t*	*df*	*p*	*d*
**Patient Demographics**								
Age	years	49.91 (17.59)	31.00 (8.90)	4.442	32.21 ^a^	0.000 **	1.362
ISI	points	19.64 (4.57)	2.89 (2.96)	13.664	39	0.000 **	4.348
**Sleep Characteristics**							
IntoBed	24 hr T (mins)	22:53 (74.59)	22:58 (43.68)	−0.294	34.62 ^a^	0.778	0.091
Bedtime	24 hr T (mins)	23:09 (68.84)	23:21 (42.81)	−0.735	35.67 ^a^	0.467	0.226
SOL	mins	34.41 (31.00)	13.13 (8.09)	3.100	24.27 ^a^	0.005 *	0.940
WASO	mins	56.02 (55.92)	7.32 (5.41)	4.062	21.48 ^a^	0.009 *	1.226
WakeT	24 hr T (mins)	06:56 (73.56)	07:24 (42.89)	−1.554	34.54 ^a^	0.129	0.478
OutBed	24 hr T (mins)	07:50 (59.89)	07:45 (47.01)	0.298	39	0.767	0.094
TIB	hrs (mins)	8.57 (85.01)	8.44 (45.19)	0.604	32.33 ^a^	0.550	0.185
TST	hrs (mins)	6.02 (119.07)	7.42 (34.75)	−3.749	25.06 ^a^	0.001 **	1.137
SE	%	69.59 (17.79)	92.32 (2.87)	−5.903	22.26 ^a^	0.000 **	1.783

^a^ Levene’s test for equality of variances ≥0.05 therefore adjusted statistics are reported, * *p* ≤ 0.05, ** *p* ≤ 0.001. SD = standard deviation, ISI = Insomnia Severity Index, IntoBed = time into bed, Bedtime = attempt to sleep time, SOL = sleep onset latency, WASO = wake time after sleep onset, WakeT = final wake up time, OutBed = time out of bed, TIB = time in bed, TST = total sleep time, SE = sleep efficiency.

**Table 2 brainsci-11-00219-t002:** Negative sleep impact scores for all 12 SIAD activities between insomnia and control groups.

Activity Q#.	I-C No.	Insomnia Mean (SD)	Control Mean (SD)	*t*	*df*	*p*	*d*
1. Exercised	17–17	1.77 (1.05)	0.32 (0.56)	5.027	24.32 ^a^	0.000 **	1.770
2. Groomed	22–19	1.41 (1.14)	0.10 (0.23)	5.234	22.89 ^a^	0.000 **	1.583
3. Health Tasks	15–15	0.99 (0.83)	0.10 (0.29)	3.924	17.42 ^a^	0.001 **	1.433
4. Ate Meals	22–19	1.15 (0.99)	0.11 (0.24)	4.781	23.83 ^a^	0.000 **	1.448
5. Work/Volunteered	18–16	1.82 (0.97)	0.37 (0.53)	5.486	26.94 ^a^	0.000 **	1.855
6. School/Studied	4–13	2.14 (1.44)	0.43 (0.63)	2.308	3.36 ^a^	0.095	0.539
7. Cared for Children	9–6	1.59 (0.91)	0.51 (0.77)	2.377	13	0.033 *	1.278
8. Housework	20–17	1.64 (0.99)	0.19 (0.35)	6.129	24.51 ^a^	0.000 **	1.958
9. Non-Contact Social	22–19	1.47 (0.92)	0.18 (0.33)	6.100	27.04 ^a^	0.000 **	1.856
10. Contact Social	22–19	1.64 (0.84)	0.22 (0.41)	6.991	31.53 ^a^	0.000 **	2.140
11. Sed. Leisure	22–19	1.76 (0.99)	0.28 (0.43)	6.308	29.69 ^a^	0.000 **	1.926
12. Non-Sed. Leisure	18–17	1.88 (1.06)	0.14 (0.27)	6.764	19.28 ^a^	0.000 **	2.229

^a^ Levene’s test for equality of variances ≥ 0.05 therefore adjusted statistics are reported, * *p* ≤ 0.05, ** *p* ≤ 0.001. I-C No. = insomnia-control groups number of participators, SD = standard deviation.

**Table 3 brainsci-11-00219-t003:** Effort scores for all 12 SIAD activities between insomnia and control groups.

Activity Q#.	I-C No.	Insomnia Mean (SD)	Control Mean (SD)	*t*	*df*	*p*	*d*
1. Exercised	17–17	2.26 (0.88)	1.56 (1.17)	1.973	32	0.057	0.677
2. Groomed	22–19	1.27 (0.86)	0.58 (0.83)	2.630	39	0.012 *	0.825
3. Health Tasks	15–15	0.80 (0.55)	0.52 (0.91)	1.049	28	0.303	0.383
4. Ate Meals	22–19	0.99 (0.82)	0.66 (0.81)	1.289	39	0.205	0.404
5. Work/Volunteered	18–16	2.24 (0.80)	1.45 (1.15)	2.338	32	0.026 *	0.795
6. School/Studied	4–13	3.01 (0.39)	1.50 (1.20)	3.931	14.75 ^a^	0.001 **	1.697
7. Cared for Children	9–6	2.03 (0.72)	1.01 (0.96)	2.341	13	0.036 *	1.194
8. Housework	20–17	1.92 (0.95)	1.01 (0.84)	3.043	35	0.004 *	1.009
9. Non-Contact Social	22–19	1.42 (0.67)	0.53 (0.63)	4.378	39	0.000 **	1.374
10. Contact Social	22–19	1.77 (0.62)	0.86 (0.93)	3.609	30.53 ^a^	0.001 **	1.146
11. Sed. Leisure	22–19	1.68 (0.78)	0.58 (0.74)	4.592	39	0.000 **	1.441
12. Non-Sed. Leisure	18–17	1.95 (0.95)	0.76 (0.76)	4.046	33	0.000 **	1.373

^a^ Levene’s test for equality of variances ≥ 0.05 therefore adjusted statistics are reported, * *p* ≤ 0.05, ** *p* ≤ 0.001. I-C No. = insomnia-control groups number of participators, SD = standard deviation.

## Data Availability

Data for this project can be found at https://github.com/kelseylbickley/SIAD-Pilot-Paper-2021.

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
