# Peer review of "The Sleep Impact on Activity Diary (SIAD): A Novel Assessment of Daytime Functioning in Insomnia"

_brainsci, 2021, doi:10.3390/brainsci11020219_

Round 1
Reviewer 1 Report
Overall, this is a mostly well-written and interesting manuscript. With further validation, it could be a useful tool to guide clinicians in the diagnosis and treatment of insomnia patients, and provide a useful measuring stick for determining the effectiveness of treatment.
There are several minor adjustments to phrasing that are required, and some examples are provided below. Authors are encouraged to check the grammar and syntax carefully, as there are a number of instances in which the syntax is awkward or inappropriate for scientific reporting.
The discussion requires more extensive review by the authors to better synthesize the findings with current theory and the literature on daily functioning in insomnia.
Minor adjustments:
Line 20: The syntax of this sentence is awkward. Please consider revising.
Line 34: Please remove “ – “ and replace with a comma
Line 41: Please replace “them” with more explicit phrasing, e.g. to target these nuanced impairments.
Line 42: Please adjust “surround” to surrounding
Line 50: Please rephrase, replacing the word “their”, e.g. these questionnaires relate…
Line 65: Please adjust “deficit reports” to deficits reported
Line 96: It would be useful to provide the range of overall global scores possible for the SIAD
Line 208: The syntax of the following is awkward, please rephrase: …”almost the exact number same of activities…”
Line 255: Changed “attempt” to attempted
Major points:
Hypotheses:
What do the authors mean by “practically”? It would be useful to the reader to have this defined in the context of the SIAD and how it will be measured.
Methods:
The authors should clearly describe how the 12 activities were determined. Authors should also include a justification for weighting each activity equally, as the tasks presented appear to range in complexity and the amount of effort required in general. This is less important for comparison between subjects, as it is relative to each individual. It may, however, have implications for the assessment of individuals prior to and after treatment, as suggested by the authors as a possible future application of the tool. For example, if a patient had only performed grooming, exercise and eating meals on the assessment prior to treatment, but had completed many more tasks, or tasks of greater complexity, before the assessment after treatment, could this impact how much effort was required for each, or subsequent, tasks, thus making pre and post global scores incomparable?
Results:
Although this study was primarily focused on determining participant perception of impairment, a comparison of daily objective sleep and global SIAD scores may further validate the findings, particularly given the significant age difference between groups. This could be a correlation between group means for sleep measures and group means for global SIAD scores for each day. This may, in part, also help address on of the limitations, in which authors state they were not able to infer whether the perceptions were related to objective or subjective performance reductions.
Discussion:
Although this is a new measurement tool and has not been implemented in previous studies, the discussion requires further examination of the findings in the context of previous research. There is limited evidence for a number of the statements made in the first and second paragraphs.
Overall, the discussion does not flow well and the findings are not well linked with any standing theoretical constructs that might help strengthen the interpretation of the findings.
Authors are encouraged to review the discussion and relate the findings to other measures of daily functioning, such as sleep related impairments from the PROMIS
The following is one example where evidence that other areas of functioning (e.g. cognitive function) show compensatory mechanisms would be helpful for validating your argument.
it is possible that the insomnia group can use more effort as an adaptive behaviour to overcome the impacts of sleep on daytime activities.
Limitations:
The fact that controls were not age matched has not been adequately addressed in the limitations section. Whilst the authors have explained that they were a convenience sample, this is a potential methodological flaw, as there may be different levels of baseline effort for participation in daily activities between older and younger people, generally. Similarly, older adults may feel greater impairment with relatively smaller sleep impact.
The following statement is a triviality and not really relevant to the findings of the study. This is unnecessary, particularly with a more thorough discussion of the lack of age-matched controls.
However, we also attempt to recruit insomnia participants from the university with limited success. We could speculate that younger individuals with insomnia may be less likely to participate in tertiary education due to their sleep problem, but this would require further investigation.
Impact and Future Directions:
Line 259: Fix spelling of the word future in the section title
This section should include a statement regarding further validation of the SIAD through comparisons with other daily function assessment tools and even objective measures of activity performance, and how this kind of validation might be undertaken.
Author Response
Reviewer 1.
Overall, this is a mostly well-written and interesting manuscript. With further validation, it could be a useful tool to guide clinicians in the diagnosis and treatment of insomnia patients, and provide a useful measuring stick for determining the effectiveness of treatment.
We thank the reviewer for the positive comments and are pleased the significance of the study was appreciated.
There are several minor adjustments to phrasing that are required, and some examples are provided below. Authors are encouraged to check the grammar and syntax carefully, as there are a number of instances in which the syntax is awkward or inappropriate for scientific reporting.
Much of the introduction and discussion has been re-written to address these general comments.
The discussion requires more extensive review by the authors to better synthesize the findings with current theory and the literature on daily functioning in insomnia.
References have been added (e.g. 10, 20,21) to the discussion to increase the integration with past literature.
Line 20: The syntax of this sentence is awkward. Please consider revising.
This long sentence was split in two and reads more easily now.
Line 34: Please remove " - " and replace with a comma
Thank you, done.
Line 41: Please replace "them" with more explicit phrasing, e.g. to target these nuanced impairments.
This suggestion has been accommodated in a re-write of this section of introduction.
Line 42: Please adjust "surround" to surrounding
This suggestion has been accommodated in a re-write of this section of introduction.
Line 50: Please rephrase, replacing the word "their", e.g. these questionnaires relate...
This suggestion has been accommodated in a re-write of this section of introduction.
Line 65: Please adjust "deficitreports" to deficits reported
Done.
Line 96: It would be useful to provide the range of overall global scores possible for the SIAD.
We were uncertain about the meaning of “global scores”. Section 2.2.2 states the range of total activity scores from 0 to 12. Impact and effort scores range possibly from 0 to 4. Figures 2 and 3 violin plots show the distribution of the average number of activities, impact, and effort scores for all participants in the two groups.
Line 208: The syntax of the following is awkward, please rephrase: ... "almost the exact number same of activities ... "
This section of the discussion has been re-written and hopefully captures the intent of this comment.
Line 255: Changed "attempt" to attempted
This occurred in a now re-written section of the discussion.
Major Points
Hypotheses:
What do the authors mean by "practically"? It would be useful to the reader to have this defined in the context of the SIAD and how it will be measured.
The hypotheses are re-written to be more specific. “we expect that the insomnia group will report engagement in fewer activities than good sleepers and for the activities in which they do engage they will report more negative impact from their prior sleep and greater perceived effort associated with engagement. In addition, the greater ecological relevance of the SIAD as a more comprehensive measure of the daytime effects of insomnia are elaborated in the re-written discussion.
Methods:
The authors should clearly describe how the 12 activities were determined.
This was expanded in section 2.2.2 of the methods and a reiteration in the discussion, “extends the information from mood and alertness feelings to a full range of common daily activities. Basic activities of daily living which are often covered by existing HRQoL questionnaires, describe basic tasks including activities like eating, bathing, and leaving the house. These are often disturbed in physically impaired populations, e.g., individuals who are unable to feed themselves due to mobility paralysis [21]. The SIAD includes these basic activities in addition to more complex daily activities such as remote and face-to-face social contact. This allows us to measure the more subtle effects of insomnia disorder on the most commonly engaged activities across the total age span.”
Authors should also include a justification for weighting each activity equally, as the tasks presented appear to range in complexity and the amount of effort required in general. This is less important for comparison between subjects, as it is relative to each individual. It may, however, have implications for the assessment of individuals prior to and after treatment, as suggested by the authors as a possible future application of the tool. For example, if a patient had only performed grooming, exercise and eating meals on the assessment prior to treatm but had completed many more tasks, or tasks of greater complexity, before the assessment after treatment, could this impact how much effort was required for each, or subsequent, tasks, thus making pre and post global scores incomparable?
Thank you for these interesting observations and suggestions.
Although these possibilities would be true for “global” scores, there is the capability of comparing the shift from some activities to others and possibly decreasing negative impact and effort from some activities but not others. The detailed analysis of the impact of treatment on specific activities is possible with the SIAD and opens a wealth of information for clinical researchers. This has been brought into the discussion in the section on “Strengths and Limitations”.
Results:
Although this study was primarily focused on determining participant perception of impairment, a comparison of daily objective sleep and global SIAD scores may further validate the findings, particularly given the significant age difference between groups.
Thank you for this good suggestion, such suggestions are now included in “Future directions”.
Discussion
Overall, the discussion does not flow well and the findings are not well linked with any standing theoretical constructs that might help strengthen the interpretation of the findings.
The discussion has been largely re-written to address many of the comments. It now reads more coherently and highlights the strengths and limitations of the study.
Limitations:
The fact that controls were not age matched has not been adequately addressed in the limitations section. Whilst the authors have explained that they were a convenience sample, this is a potential methodological flaw, as there may be different levels of baseline effort for participation in daily activities between older and younger people, generally. Similarly, older adults may feel greater impairment with relatively smaller sleep impact.
Thank you for this important comment. This has been addressed in the second paragraph in the Limitations section. “A major limitation of this pilot study was the demographic differences between the groups with the insomnia patients being older and less likely to be engaged in educational activities.. It could be argued that perceived effort to engage in daily activities increases with age and thus confounded the effort measure. In addition there is the possibility that age increases the perceived negative impact of poor sleep on these activities. Although the control group was a sample “of convenience”, future comparisons between good sleepers and insomnia patients would benefit from a more careful matching of demographics between groups.”
Impact and Future Directions:
Line 259: Fix spelling of the word future in the section title
This section should include a statement regarding further validation of the SIAD through comparisons with other daily function assessment tools and even objective measures of activity performance, and how this kind of validation might be undertaken.
Thank you for these useful suggestions. They are now addressed in the Future Directions section.
Reviewer 2 Report
Bickley and al created and tested a new method to assess subjective daytime functioning and the impact of prior night-time sleep on the participation in 12 daytime activities in insomnia and healthy sleepers, using daily questionnaire during 14 days.
A larger sample size would be needed to fully investigate the current results but the current pool of participants shows promising results and the SIAD could be a useful tools in the future to assess subjective vs objective daytime impairments.
I have some few comments:
- Introduction explains well the current literature and debate on subjective versus objective daytime impairments as well as the need for diary format rather than using one-time questionnaire on the past week.
- I would be careful in the assumption that individuals with insomnia only perform worse in objective memory, perception, and problem-solving tasks but not in attention when the matter is still in debate and dependent on so many factors (age, sample size, tasks, methods, circadian, etc etc). As you write latter, objective tasks might not be sensitive enough.
- More than not sensitive enough, they might not meet ecological enough compared to the activities your subjects subjectively assessed which is a great plus for your study
- L58: “Regarding the inclusion of effort assessments, several authors have proposed that insomnia patients may perform at normal levels in short-duration cognitive tasks by putting in more effort than controls. We propose that they would do the 60 same to participate in daytime activities.” Not sure how that relate to your assessment, this mostly refer to performance degradation with time.
- They subjectively averaged the amount of time for each activity, did you ask a question for confidence?
- Although I understand the importance to keep it simple, I was wondering why not asking how many times they performed the activity and/or when? It would have been interesting to see if there was some circadian component (less effort in the morning vs evening for example).
- Did they report if the day was a day off or a work day? WE vs week?
- You report number of subjects with complete and incomplete data but how did you handle missing data? Did you perform a mean-centered or raw mean? Often sleep diaries measures (especially in insomnia) are not normally distributed, how did you perform your stat? transformation or non-parametrical?
- With such a nice large dataset, it seems a waste to not go further and report day-by-day variance rather than a simple average. Or even perform similar analyses on standard deviation as we could hypothesized more day-by-day change in insomnia
- You have Age difference between your group, why not using Age as a covariate in an ANCOVA?
- Why not use the sleep diaries in conjunction with the SIAD? Is there any correlation between subjective TST and subjective effort/activity count/negative sleep impact? Or link to WASO, SOL etc etc? Please report the lack of association if no link
Author Response
Reviewer 2 comments, replies are underlined.
Introduction explains well the current literature and debate on subjective versus objective daytime impairments as well as the need for diary format rather than using one-time questionnaire on the past week.
The authors thank the reviewer for this positive comment.
I would be careful in the assumption that individuals with insomnia only perform worse in objective memory, perception, and problem-solving tasks but not in attention when the matter is still in debate and dependent on so many factors (age, sample size, tasks, methods, circadian, etc etc). As you write latter, objective tasks might not be sensitive enough.
We agree and have re-written part of the introduction to reflect the uncertainty about objective performance differences in insomnia to raise the possibility that the laboratory experimental studies may not capture the extent of the impact of insomnia on normal daily functioning which the SIAD attempts to elucidate in more detail.
More than not sensitive enough (laboratory studies), they might not meet ecological enough compared to the activities your subjects subjectively assessed which is a great plus for your study.
Thank you for your observation.
L58: "Regarding the inclusion of effort assessments, several authors have proposed that insomnia patients may perform at normal levels in short-duration cognitive tasks by putting in more effort than controls. We propose that they would do the 60 same to participate in daytime activities." Not sure how that relate to your assessment, this mostly refer to performance degradation with time.
You are correct, we implied a time factor here but now relate it to the normal everyday circumstances, (Insomnia patients may), . “. . not be able to sustain this effort throughout more extended normal daily conditions.”
Although I understand the importance to keep it simple, I was wondering why not asking how many times they performed the activity and/or when?
This is always a salient comment in the development of a questionnaire administered daily over a period of time. This issue is discussed in the section on Limitations, “Another potential limitation of the present SIAD tool is that it does not go far enough to detail the number of times, energy expended, or amount of time spent in each activity. An activity type engaged in once for a short time (e.g. single, brief walking exercise) counts the same as an activity engaged in several times for much longer (e.g. a few bouts of heavy exercise over longer period of total time). However, the latter amount of detail would have required considerably more time to document in a questionnaire and most probably have reduced compliance and increased the percentage of missing data. We tried to strike a reasonable compromise between capturing additional information of daily activities without requiring so much information to seriously degrade compliance with increasing missing data. Although there was some missing data in the present study, its small percentage (~5%) suggested to us that the SIAD struck the right balance between maximizing information while minimizing onerousness.”
Did they report if the day was a day off or a work day? WE vs
Thank you for these important points, these are raised in the Future Directions section.
You report number of subjects with complete and incomplete data but how did you handle missing data? Did you perform a mean-centered or raw mean?
Thank you for the comment, this is almost always an issue in studies requiring a decision on the amount of missing data and whether it warrants an attempt to statistically compensate for that missing data, always a process invoking assumptions and estimations with a degree of intrinsic error. We took the decision in our case that statistical compensation was not indicated with a low rate of missing data (5.6%) and detailed this in the “Missing data” section 2.4.1.
Often sleep diaries measures (especially in insomnia) are not normally distributed, how did you perform your stat? transformation or non-parametrical?
We agree with your comments about the distribution of sleep diary measures, especially SOL and WASO in the insomnia group as shown in Table 1. However, the independent variable was grouping into Insomnia and good sleepers on the basis of the ISI measure that showed such a large effect size (d=4.4) with relatively small SDs in each group that we felt confident that the robust t-test on raw data was adequate to confirm a large group difference in reported sleep quality. The t-tests on SOL, WASO, TST, and SE we feel were adequate to confirm this large group difference in sleep.
With such a nice large dataset, it seems a waste to not go further and report day-by-day variance rather than a simple average. Or even perform similar analyses on standard deviation as we could hypothesized more day-by-day change in insomnia.
We agree that such an analysis as well as correlation between sleep and SIAD variables would be of great interest but in an expanded study (participant numbers and time period) that we feel was beyond the scope of the present study as we propose in the Future Directions section, “Although it was beyond the scope of this study with only 14 days of data, it would be important to investigate intra-individual day-to-day variation in sleep measures (both from the social variation between weekday and weekend and in spontaneous variation) with variation in number of activities, negative impact on activities, and required effort in activities.”
Round 2
Reviewer 2 Report
Authors responded adequately to my comments and I support publication!